# Molecular detection of dugbe orthonairovirus in cattle and their infesting ticks (*Amblyomma* and *Rhipicephalus* (*Boophilus*)) in Nigeria

**Oluwafemi Babatunde Daodu**[1], **Albert Eisenbarth**[2,3], **Ansgar Schulz**[2], **Julia Hartlaub**[2], **James Olukayode Olopade**[4], **Daniel Oladimeji Oluwayelu**[5,6], **Martin H. Groschup**[2]*

1 Department of Veterinary Microbiology, Faculty of Veterinary Medicine, University of Ilorin, Ilorin, Nigeria, 2 Institute of Novel and Emerging Infectious Diseases, Friedrich-Loeffler-Institut, Greifswald–Insel Riems, Germany, 3 Bundeswehr Hospital Hamburg, Branch Tropical Microbiology and Entomology, Hamburg, Germany, 4 Department of Veterinary Anatomy, Faculty of Veterinary Medicine, University of Ibadan, Ibadan, Nigeria, 5 Department of Veterinary Microbiology, Faculty of Veterinary Medicine, University of Ibadan, Ibadan, Nigeria, 6 Center for Control and Prevention of Zoonoses, Faculty of Veterinary Medicine, University of Ibadan, Ibadan, Nigeria

* martin.groschup@fli.de

**Data Availability Statement:** The data supporting the results of this study are available upon request from the Prof. Dr. Jonas Schmidt-Chanasit,

## Abstract

Dugbe orthonairovirus (DUGV), a tick-borne zoonotic arbovirus, was first isolated in 1964 in Nigeria. For over four decades, no active surveillance was conducted to monitor the spread and genetic variation of DUGV. This study detected and genetically characterized DUGV circulating in cattle and their infesting ticks (*Amblyomma* and *Rhipicephalus* (*Boophilus*)) in Kwara State, North-Central Nigeria. Blood and or ticks were collected from 1051 cattle at 31 sampling sites (abattoirs and farms) across 10 local government areas of the State. DUGV detection was carried out by RT-qPCR, and positive samples sequenced and phylogenetically analysed. A total of 11824 ticks, mostly *A. variegatum* (36.0%) and *R.* (*B.*) *microplus* (63.9%), were obtained with mean tick burden of 12 ticks/cattle. Thirty-four (32 *A. variegatum* and two *R.* (*B.*) *microplus*) of 4644 examined ticks were DUGV-positive, whereas all of the cattle sera tested negative for DUGV genome. Whole genome sequence (S, M and L segments) and phylogenetic analyses indicate that the positive samples shared up to 99.88% nucleotide identity with and clustered around the Nigerian DUGV prototype strain IbAr 1792. Hence, DUGV with high similarity to the previously characterised strain has been detected in Nigeria. To our knowledge, this is the first report of DUGV in North-Central Nigeria and the most recent information after its last surveillance in 1974.

## Author summary

More than half a century after the discovery of a new, potentially zoonotic virus transmitted by ticks, the prevalence of Dugbe virus in cattle and its tick vector was investigated in Nigeria. The survey took place in Kwara State on the border with the Republic of Benin over a period of one year. More than 1,000 cattle were examined across the state and nearly 12,000 ticks were collected and identified under the microscope. Using established

Bernhard Nocht Institute for Tropical Medicine, Bernhard-Nocht-Strasse 74, 20359 Hamburg, schmidt-chanasit@bni-hamburg.de. Due to the privacy of the research participants, especially the contributing pastoralists and herdsmen, individual data are not publicly accessible.

**Funding:** M.H.G and J.O.O. received funding for this study from the Alexander-von-Humboldt Foundation for financial support of this work. The funders had no role in study design, data collection and analysis, decision to publish, or preparation of the manuscript.

**Competing interests:** The authors have declared that no competing interests exist.

diagnostic protocols for molecular detection of Dugbe virus by quantitative PCR, we found no virus in the cattle population, but about 0.7% (n = 34) of the ticks were carriers of the virus. The overwhelming majority of ticks collected from cattle belonged to only two species: *Amblyomma variegatum* (tropical cattle tick) and *Rhipicephalus* (*Boophilus*) *microplus* (Asian blue-footed tick). In addition, most Dugbe virus infections were found in A. variegatum, and only two *R.* (*B.*) *microplus* were also infected. Although there is no clear evidence that *A. variegatum* is the main vector of Dugbe virus, our whole virus genome data showed a remarkably high similarity with Dugbe virus first isolated in Nigeria in 1964 in the same tick species.

## Introduction

Dugbe orthonairovirus (DUGV) was first isolated in Nigeria from *Amblyomma variegatum* ticks collected from cattle at the Dugbe abattoir in Ibadan in 1964 [1]. It is a member of the genus O*rthonairovirus*, family Nairoviridae, to which highly pathogenic viruses such as Crimean-Congo hemorrhagic fever orthonairovirus (CCHFV) and Nairobi sheep disease orthonairovirus (NSDV) belong [2]. Like other Orthonairoviruses, DUGV is an enveloped virus with tripartite single-stranded, negative-sense RNA segments designated as L, M and S segments that encode for the RNA-dependent RNA polymerase, viral glycoproteins (Gn and Gc) and the nucleocapsid protein (N), respectively [3]. Related DUGV serogroup viruses have been identified in several countries such as Ganjam virus in India and Sri Lanka, and Nairobi sheep disease orthonairovirus and Kupe virus in Kenya [4,5].

The tropical bont tick (*Amblyomma variegatum*) has been described as the primary competent vector of DUGV since the virus can be transmitted trans-stadially and trans-ovarially among this tick species [6,7]. DUGV has also been recovered from other tick species (*Rhipicephalus* (*Boophilus*) and *Hyalomma* species), cattle and humans [8]. Subsequently, it has been isolated in tick-borne virus surveillance [9,10,11] and serological monitoring studies [12,13] conducted in different African countries. Hence, evidence exists that DUGV is distributed in at least 13 African countries including Nigeria, Senegal, Cameroon, Central African Republic, Ethiopia, Sudan, Uganda, Chad, Kenya, Egypt, South Africa, Guinea and Ghana [1,9,14–16].

Ticks, livestock (cattle, goats, sheep and camels), (small) wild mammals, monkeys and birds have been proposed to maintain DUGV in the environment. Humans are known to be susceptible to DUGV [17,18] which has been categorized as a BSL 3 pathogen [19]. Symptoms in humans, when they occur, are described as a mild febrile illness [12]. Apart from the initial studies conducted on DUGV in the 1960s-1970s in Nigeria [8,12,20], no additional work on the virus has been reported in the country for about four decades. In addition, previous research did not investigate the genetic characteristics of circulating DUGV in Nigeria. This study was therefore designed to detect and characterize DUGV strains circulating among cattle as well as *Amblyomma* and *Rhipicephalus* (*Boophilus*) ticks feeding on them in Kwara State, Nigeria.

## Materials and methods

### Ethical statement

Samples were collected according to fundamental ethical principles for diagnostic purposes in context of national surveillance studies. Animal sera and ticks were collected as approved by the Animal Care and Use Research Ethics Committee (ACUREC), University of Ibadan, Ibadan, Nigeria (UI-ACUREC/18/0143).

### Study area and sampling sites

The study was conducted in Kwara State of Nigeria which lies on latitude 8˚ to 10˚ N and longitude 3˚ to 6˚ E with a total land mass of 77,865 hectares. Based on Land-Sat image, the State's land mass is covered by 47.8% forest vegetation, 35% savannah vegetation, 16.7% built-up areas, and 0.4% water bodies [21]. Thirty-one sampling sites in 10 local government areas (LGAs) of the State were arbitrarily selected. Samples were obtained between February 2017 to March 2018 from pastoralists' settlements and abattoirs from where 5% of the cattle slaughtered were sampled every 2–3 weeks. The pastoralists' settlements were visited once while abattoirs were visited at 1–2 week intervals for sampling.

### Blood sample and tick collection

Five milliliters of blood was obtained through the jugular vein from each of 904 cattle under aseptic conditions. The samples were dispensed into sterile plain tubes (without anticoagulant), placed in a slanting position and transported to the laboratory under cold chain conditions. Subsequently, they were centrifuged at 1,500 rpm for 10 minutes and the sera were separated into sterile cryovials before being stored at -20˚C until used. Ticks on the brisket, udders and perineum of the bled cattle (n = 904) and unbled cattle (n = 147) were arbitrarily collected into sterile tubes and immediately transported to the laboratory under cold chain conditions. The ticks were rapidly washed with sterile distilled water to remove any dirt. They were then placed in tubes containing 70% ethanol and stored at -20˚C until further use. Information obtained on each sampled cattle such as sex, location and management system were also recorded.

### Identification of tick species

The ticks were sorted according to their host animal and identified to sex, species and developmental stage using established morphological keys [22]. Additionally, representative samples of the tick species were subjected to molecular identification by genetic barcoding for confirmation of the species [23]. Positive amplicons were confirmed by 1.2% agarose gel electrophoresis and prepared for Sanger sequencing (Eurofins Genomics Germany GmbH, Ebersberg, Germany). Manual quality control and nBLAST alignments with entries from GenBank (NCBI, Bethesda, USA) were performed with the Geneious bioinformatics software (Biomatters, Auckland, New Zealand).

### *Amblyomma* and *Rhipicephalus* (*Boophilus*) tick selection and homogenization

A total of ≤ five ticks/cattle were selected based on tick species, animal origin, location and season of sampling for subsequent pooling. Selected ticks were placed in 2 ml Eppendorf tube containing 300 μl AVL buffer with carrier RNA (Qiagen, Hilden, Germany) and a steel bead (3 mm for small and 5 mm for big ticks) (Isometall, Pleidelsheim, Germany). Fully engorged ticks which did not fit into the tube were excluded.

The ticks were homogenized using a Tissue Lyser (TissueLyser II, Qiagen, Hilden, Germany) at 30 Hz for 3 minutes and the homogenate was centrifuged at 10,000 rpm for 10 min. The supernatant was aspirated and dispensed into a new tube. Then, a pool of supernatants was made by taking at least 40 μl each from five selected ticks to obtain a total of 200 μl. However, if < five ticks were obtained from one animal, the volume of supernatant taken was more than 40 μl to ensure that the pooled volume was 200 μl. The pooled supernatants were heat-inactivated at 70˚C for 10 minutes. Each 200 μl pooled supernatant and the remaining individual supernatants were then stored at -20˚C until extracted for RNA.

### Viral RNA extraction and molecular detection

RNA was extracted using MS2 as extraction control from cattle sera and tick supernatants using the Nucleomag Vet kit (Machery-Nagel, Düren, Germany) according to the manufacturer's instructions and KingFisher Flex Purification System (ThermoFisher Scientific, Waltham, USA). Real-time reverse transcriptase-polymerase chain reaction on the S segment was carried out based on the method of Hartlaub et al. [24] using a CFX96 Real-Time PCR Detection System (Bio-Rad Laboratories, Hercules, USA). Subsequently, individual ticks that constituted the positive tick pools were individually extracted and re-examined for DUGV.

### Sequencing and phylogenetic analysis

Selected positive samples were sequenced by Sanger sequencing (Eurofins Scientific, Luxemburg) using the amplicon (196 bp) of the RT-qPCR (without probe) described in the section above. Afterwards, the obtained sequences were analysed in Geneious prime (Version 2021.0.1) and compared using the nBLAST tool (https://blast.ncbi.nlm.nih.gov/Blast.cgi). In order to obtain a more meaningful phylogenetic conclusion, the S and M segment of one sample was sequenced using nanopore sequencing techniques. Base calling, demultiplexing and adapter trimming were performed using Guppy version 3.2.10 in the MK1C sequencer (Oxford Nanopore Technologies, Oxford, United Kingdom). Fastq reads were used for alignment against redundant databases in using KMA [25].

Fourteen published S segment sequences (nine Dugbe orthonairovirus, one Kupe virus, two Nairobi sheep disease orthonairovirus and two Crimean-Congo hemorrhagic fever orthonairovirus), fifteen M segment sequences (nine Dugbe orthonairovirus, four Kupe orthonairovirus, one Nairobi Sheep disease virus (Ganjam virus) and one Crimean-Congo haemorrhagic fever orthonairovirus) and fifteen L segments sequences (eight Dugbe orthonairovirus, four Kupe orthonairovirus, two Crimean-Congo haemorrhagic fever orthonairovirus and one Hazara orthonairovirus) were downloaded from NCBI GenBank database. Then, multiple alignments for sequences of each of our S, M and L segments were carried out using MAFFT v.7 software [26] while phylogenetic analysis was conducted with MEGA X [27] using the maximum likelihood method and Tamura-Nei model. The phylogenetic tree was inferred from 1,000 replicates based on the general time-reversible substitution model with gamma distribution.

### Statistical analysis

Data obtained from this study were applied into the Statistical Package for Social Sciences software, version 22 (SPSS, Illinois, USA). Descriptive and inferential statistics were used to analyze the data, and the paired t-test and Chi- squared test were used to determine the level of significance with p-value set at $\leq 0.05$.

## Results

### Blood sample and tick collection

From 1,051 cattle, 11,826 ticks comprising 4261 *Amblyomma variegatum*, 7557 *Rhipicephalus* (*Boophilus*) *microplus*, four *R.* (*B.*) *annulatus*, one *R.* (*B.*) *geigyi* and three *R.* (*B.*) *decoloratus* were collected. Blood was obtained from 904 cattle. Among *A. variegatum* obtained, adult males (2,379/4,261, 55.8%) were highest followed by adult females (22.6%), nymphs (20.9%) and larvae (0.6%) (Table 1). While *R.* (*B.*) *microplus* (7,557/7,565; 99.9%) was highest among *Rhipicephalus* species, adult females were the most abundant tick stage for *R.* (*B.*) *microplus* (5,854/7,557; 77.5%) and the larval stage was the least (7/7,557; 0.1%). *Rhipicephalus* (*Boophilus*) species

**Table 1. Distribution of ticks infesting cattle in Kwara State, Nigeria.**

| Tick species | Tick developmental stage | Tick count (%) | Mean ± SE | Std. Dev |
|---|---|---|---|---|
| *A. variegatum* (n = 4261) | Adult male | 2,379 (55.83) | 2.26 ± 0.12 | 3.97 |
| | Adult female | 965 (22.65) | 0.92 ± 0.06 | 1.88 |
| | Nymph | 892 (20.93) | 0.85 ± 0.06 | 1.98 |
| | Larva | 25 (0.59) | 0.02 ± 0.01 | 0.31 |
| *R. (B.) microplus* (n = 7557) | Adult male | 1,316 (17.41) | 1.25 ± 0.09 | 2.85 |
| | Adult female | 5,854 (77.46) | 5.57 ± 0.29 | 9.25 |
| | Nymph | 380 (5.03) | 0.36 ± 0.04 | 1.38 |
| | Larva | 7 (0.09) | 0.01 → 0.00 | 0.16 |
| *R. (B.) annulatus* (n = 4) | Adult female | 4 (100.00) | 0.00 ± 0.00 | 0.08 |
| *R. (B.) geigyi* (n = 1) | Adult male | 1 (100.00) | 0.00 ± 0.00 | 0.03 |
| *R. (B.) decoloratus* (n = 3) | Adult female | 3 (100.00) | 0.00 ± 0.00 | 0.09 |

**Key:** n = count of tick SE- Standard error of mean

constituted a higher tick burden on cattle (7 ticks per cattle) than *A. variegatum* (4 ticks per cattle). Based on *A. variegatum* tick stage, adult males had highest burden on cattle ($\geq 2$ ticks/cattle) compared with other stages of *A. variegatum*. However, adult female *R. (B.) microplus* constituted the highest burden on cattle (6 ticks/cattle). Based on local government area sampled, the results showed that higher *A. variegatum* burden exists in sampling sites at Asa LGA (14 ticks/cattle), Baruten LGA (8 ticks/cattle) and Ilorin South LGA (8 ticks/cattle) compared with other sampling sites. However, higher *R. (B.) microplus* burden exists in sampling sites in Baruten (23 ticks/cattle) and Kaiama LGAs relative to other sampling sites.

## RT-qPCR

Based on the tick selection described above, a total of 1,228 tick pools (4,644 individual ticks) were formed of which 18 pools and 1 pool were DUGV positive and doubtful respectively. Among these pools, only 34 ticks were DUGV-positive (0.7%; 34/4,644). Multiple positive ticks were found in 17 of the 18 DUGV positive pools (Table 2). These DUGV-positive ticks included 32/2,301 (1.4%) *A. variegatum* (23 adult males and 9 adult females) and 2/2,337 (0.1%) *R. (B.) microplus* (2 adult female ticks).

**Table 2. Distribution of DUGV-positive ticks in Kwara State, Niger.**

| Tick species | Total No. of ticks collected | Total No. of ticks tested | DUGV-positive (%) | Tick Stage/DUGV positive | | | | | | | |
|---|---|---|---|---|---|---|---|---|---|---|---|
| | | | | Adult Male | | Adult Female | | Nymph | | Larva | |
| | | | | Tick count | DUGV-positive (%) | Tick count | DUGV-positive (%) | Tick count | DUGV-positive (%) | Tick count | DUGV-positive (%) |
| *A. variegatum* | 4,261 | 2,301 | 32 (1.4) | 1,177 | 23 (2.0) | 578 | 9 (1.5) | 532 | 0 (0.0) | 14 | 0 (0.0) |
| *R. (B.) microplus* | 7,557 | 2,337 | 2 (0.1) | 577 | 0 (0.0) | 1,612 | 2 (0.1) | 148 | 0 (0.0) | 0 | 0 (0.0) |
| *R. (B.) annulatus* | 4 | 2 | 0 (0.0) | 0 | 0 (0.0) | 2 | 0 (0.0) | 0 | 0 (0.0) | 0 | 0 (0.0) |
| *R. (B.) geigyi* | 1 | 1 | 0 (0.0) | 1 | 0 (0.0) | 0 | 0 (0.0) | 0 | 0 (0.0) | 0 | 0 (0.0) |
| *R. (B.) decoloratus* | 3 | 3 | 0 (0.0) | 0 | 0 (0.0) | 3 | 0 (0.0) | 0 | 0 (0.0) | 0 | 0 (0.0) |
| **Total** | **11,826** | **4,644** | **34 (0.7)** | **1,755** | **23 (1.3)** | **2,195** | **11 (0.5)** | **680** | **0 (0.0)** | **14** | **0 (0.0)** |

**Key:** *A.—Amblyomma variegatum R. (B.)–Rhipicephalus (Boophilus)*

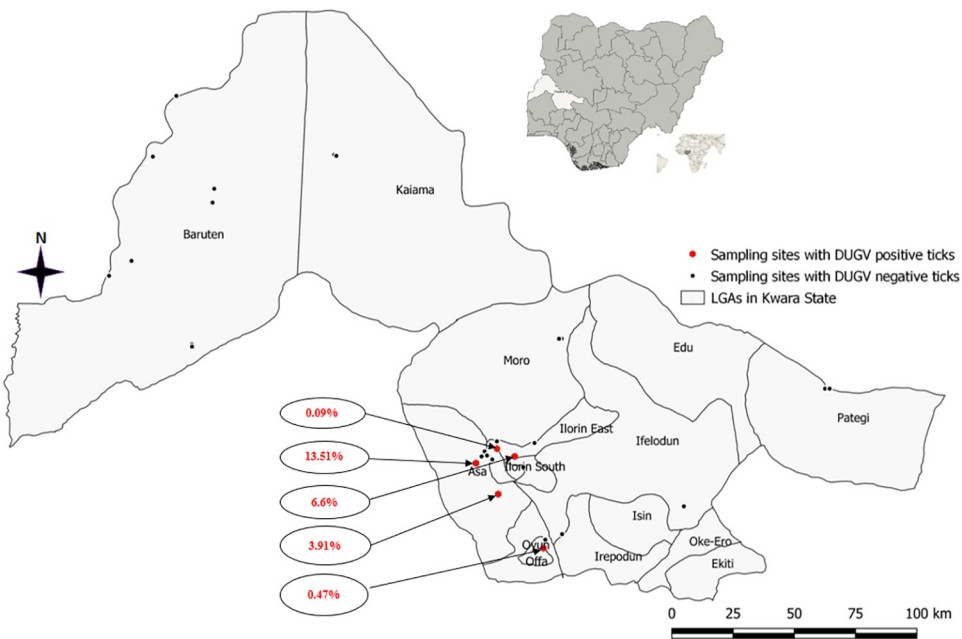

**Fig 1. Map of Kwara State of Nigeria showing sampling sites.** DUGV-positive sites in red and DUGV-negative sites in black circles. Values in red colour indicate DUGV detection rates at individual sampling sites based on RT-qPCR. The base layer of the map was created using DIVA-GIS Version 7.5 (https://www.diva-gis.org/) software.

Five out of the 31 sampling sites (located in four of the 10 LGAs selected) had DUGV-positive ticks and DUGV presence ranged from 0.09–13.51% (Fig 1; Table 3). Further, the results indicate that DUGV was detected at two sampling sites at Asa LGA (A: 15/111, 13.5%; B: 9/230, 3.9%) (Fig 1). Other LGAs where DUGV-positive ticks were detected include Ilorin East (7/1,716; 0.4%), Offa (2/423; 0.5%) and Ilorin West (1/696; 0.1%) (Fig 2).

Based on the ticks selected for DUGV detection, the result showed that more DUGV-positive *A. variegatum* were found during the dry (23/904; 2.54%) than the rainy (9/1,397; 0.64%)

**Table 3. Distribution of *Amblyomma variegatum* and *Rhipicephalus* (*Boophilus*) *microplus* infesting cattle in Kwara State based on location.**

| Location | No. of Cattle | *Amblyomma variegatum* | | | [a]*Rhipicephalus (Boophilus) microplus* | | |
|---|---|---|---|---|---|---|---|
| | | Tick count | Mean ± SE | Std. Dev | Tick count | Mean ± SE | Std. Dev |
| Ilorin West | 597 | 1,470 | 2.46 ± 0.18 | 4.51 | 2,745 | 4.60 ± 0.38 | 9.39 |
| Patigi | 15 | 27 | 1.80 ± 0.50 | 1.93 | 42 | 2.80 ± 2.26 | 8.74 |
| Ilorin East | 107 | 850 | 7.94 ± 0.66 | 6.81 | 1,300 | 12.15 ± 1.61 | 16.61 |
| Ilorin South | 41 | 237 | 5.78 ± 0.75 | 4.83 | 211 | 5.15 ± 0.81 | 5.16 |
| Offa | 76 | 222 | 2.92 ± 0.40 | 3.52 | 671 | 8.83 ± 1.01 | 8.76 |
| Oyun | 60 | 173 | 2.88 ± 0.30 | 2.34 | 336 | 5.60 ± 0.78 | 6.03 |
| Asa | 49 | 631 | 12.88 ± 1.44 | 10.05 | 315 | 6.43 ± 0.91 | 6.37 |
| Moro | 11 | 7 | 0.64 ± 0.20 | 0.67 | 66 | 6.00 ± 1.14 | 3.79 |
| Baruten | 47 | 392 | 8.34 ± 1.18 | 8.06 | 1,078 | 22.94 ± 2.50 | 17.13 |
| Kaiama | 48 | 252 | 5.25 ± 0.63 | 4.33 | 793 | 16.52 ± 2.02 | 14.00 |
| **Total** | **1,051** | **4,261** | **4.05 ± 0.18** | **5.89** | **7,557** | **7.19 ± 0.36** | **11.54** |

**Key:**

[a]Only *R.* (*B.*) *microplus* was considered SE- Standard error of mean Std. Dev- Standard deviation 95%CI- Confidence interval

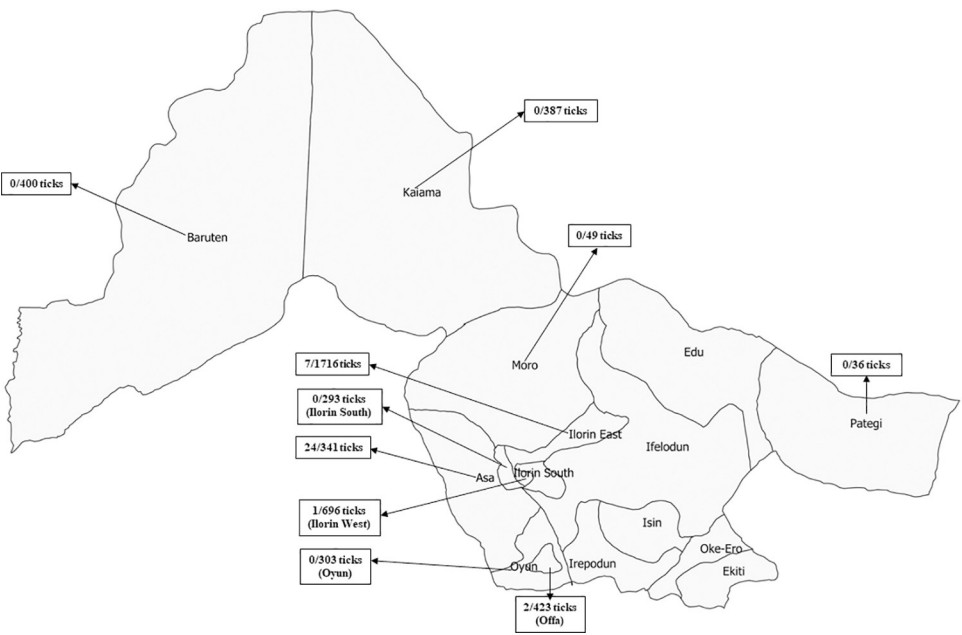

**Fig 2. Map of Kwara State of Nigeria showing local government areas with number of DUGV positive ticks against total number of ticks tested (box).** The base layer of the map was created using DIVA-GIS Version 7.5 (https://www.diva-gis.org/) software.

season (Table 4). Also, DUGV detection rate in selected *A. variegatum* was higher in pastoralists' settlements compared to abattoirs.

**Sequence and phylogenetic analysis.** Subsequently, five ticks which had high virus loads (Ct values range 20.5–26.1) were selected for sequencing. Meaningful sequencing results were available from four out of five sequenced samples. The S-segment sequence of best quality had 95.8% nucleotide sequence homology (100% coverage) with DUGV isolates IbAr 1792 from Nigeria (Accession Numbers KU925457.1 and AF434164.1), and ArD 16095 and ArD 44313 from Senegal (AF434162.1 and AF434161.1). Further, analysis of mapped nanopore sequencing reads revealed a consensus sequence of the S segment (1,673 bp), M segment (4,900 bp) and L segment (12,357 bp) making a whole genome sequence of 18,930 bp. When our sequences were compared with the published DUGV strain IbAr 1792 isolated from *A. variegatum* infesting cattle in Nigeria, analysis showed 99.88% nucleotide and 99.76% amino acid similarities in S segment, 96.01% nucleotide and 95.90% amino acid similarities in M segment, and 96.62% nucleotide and 96.31% amino acid similarities in L segment. In addition,

**Table 4. Distribution of DUGV-positive *A. variegatum* based on season, sex and location of cattle harboring ticks in Kwara State, Nigeria.**

| Features | | No of ticks collected (%) | No of ticks tested | DUGV-positive ticks (%) | OR (95% CI) | $X^2$ | p-value |
|---|---|---|---|---|---|---|---|
| *Season* | Dry | 1,402 | 904 | 23 (2.54) | 4.02 (1.85–8.74) | 13.10 | 0.0003* |
| | Rain | 2,859 | 1,397 | 9 (0.64) | r | | |
| *Animal location* | Abattoir | 1,866 | 1,162 | 3 (0.26) | r | | |
| | Farm | 2,395 | 1,139 | 29 (2.55) | 10.09 (3.07–33.24) | 20.32 | <0.0001* |
| *Sex of Cattle* | Male | 1,063 | 509 | 12 (2.36) | 2.14 (1.04–4.41) | 3.60 | 0.0579 |
| | Female | 3,198 | 1,792 | 20 (1.12) | r | | |

Key: $X^2$- Chi Square *Significant at p<0.05 r- reference value

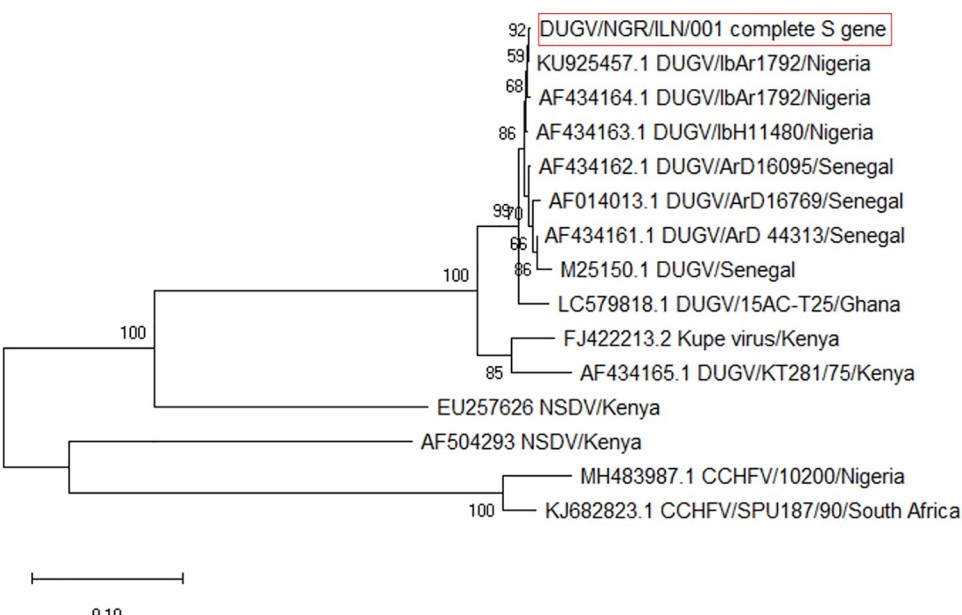

**Fig 3. Phylogenetic relationship among DUGV strain recovered in this study and reference strains based on nucleotide sequence of a 1,673 bp fragment of the S segment gene.**

phylogenetic analysis based on the sequences of this strain indicated that our S and M segments clustered with IbAr 1792 S segment (accession number: KU925457.1) (Fig 3) and the IbAr 1792 M segment (accession number: KU925456.1) (Fig 4) respectively while L segment

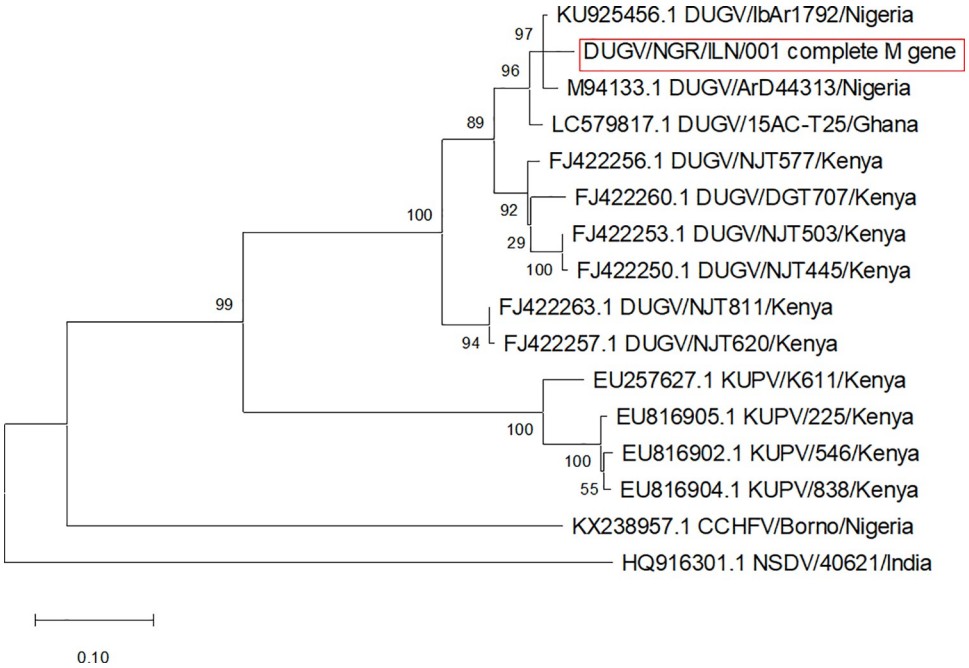

**Fig 4. Phylogenetic relationship among DUGV strain recovered in this study and reference strains based on nucleotide sequence of a 4,900 bp fragment of the M segment gene.**

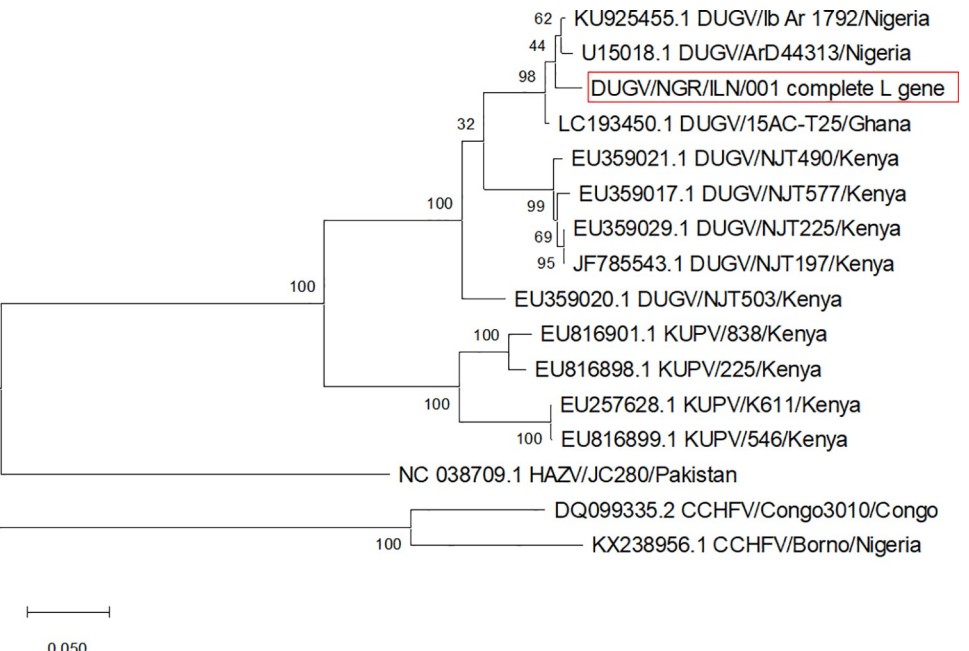

**Fig 5. Phylogenetic relationship among DUGV strain recovered in this study and reference strains based on nucleotide sequence of a 12357 bp fragment of the L segment gene.**

clustered with the same segment of IbAr 1792 (accession number: KU925455.1) and ArD 44313 (U15018.1) Dugbe orthonairovirus strains (Fig 5).

## Discussion

There are only few studies on DUGV in Nigeria which were carried out more than four decades ago when it was first discovered. Therefore, the presented study provides up-to-date information on the presence of DUGV in Nigeria. DUGV was detected in 34 ticks (0.7%) out of 4,644 ticks tested. To our knowledge, this is the first report of DUGV in North-Central Nigeria and the latest information on the virus after more than four decades of no report in Nigeria. This indicates that DUGV is currently present in the tick population in Nigeria after its first detection in 1964 [1]. The higher DUGV detection rate among *A. variegatum* compared with *Rhipicephalus* (*Boophilus*) species in this study underlines their important role in maintenance of the virus. The study showed also that among *Rhipicephalus* (*Boophilus*) ticks, only two adult female *R.* (*B.*) *microplus* (2/2,337; 0.1%) were DUGV-positive while adult males (23/1,177; 2%) and adult females (9/578; 1.5%) of *A. variegatum* tested positive. Previous studies had indicated several DUGV detection in *A. variegatum* involving all stages while few others were isolated from other ixodid ticks [4,6,7,16]. However, since the ticks were collected from cattle and are therefore considered "blood-fed", it is impossible to draw any definite conclusions on vector competence based on these data. Further, our study showed that DUGV-negative cattle could harbor DUGV-infected tick even though all the cattle sera tested negative. This could mean either that DUGV was not successfully transmitted to cattle infested by DUGV-infected tick or that blood samples were collected during the non-viremic phase. We also observed the presence of DUGV-positive and -negative ticks (of identical or different species) infesting the same cattle. This suggests that either some of the DUGV-positive ticks might have been infected from previous hosts or got infected by their present host during the viremic phase or via co-feeding.

*Amblyomma variegatum* and *Rhipicephalus* (*Boophilus*) *microplus* were the major species found on cattle at the study sites. Previous reports in Nigeria have shown varying tick species population probably due to differences in sampling location (vegetational zones), season and duration of sampling used [28–32]. Our study spanned for one year and covered two ecological zones (forest and guinea savannah) which were absent in previous reports. Further, the overall presence of 11,826 ticks on 1,051 cattle implies that on average 11 ticks were found on each animal. This burden was higher than three and six ticks per cattle reported in Enugu and Nasarawa States [28,31] but lower than 14 and 22 ticks per cattle reported in Benue [29] and Plateau States [30]. Also, our findings showed significant differences (p<0.0001) for each of the tick stages found on cattle for *A. variegatum* and *R.* (*B.*) *microplus* (Table 1).

Despite the greater abundance of *A. variegatum* and *Rhipicephalus* species during the rainy than the dry season, DUGV was more frequently detected in *A. variegatum* collected during the dry season (Table 4). This study is limited to be able to explain the reason for this discrepancy, however, a herd based DUGV study could explain the significance of season in DUGV epidemiology.

The nBLAST analyses revealed that our isolates displayed high similarity (95.8%) with DUGV IbAr 1792 strain recovered from *A. variegatum* tick collected in Nigeria during its first discovery in 1964 [1]. This suggests that the virus isolates from this study are relatively similar to the previous DUGV strain recovered in Nigeria. This was also underlined by the high similarity of the complete S and M segment sequences produced by nanopore sequencing. While the virus has already been reported in several African countries [1,9,13,14,15,18], the available sequences in the GenBank are mainly limited to Nigeria, Kenya, Senegal and Ghana [33]. Therefore, in order to obtain a better understanding of the genetic and geographical diversity of DUGV, future screenings for more virus isolates targeting the above mentioned and other countries, and the generation of more published sequences are necessary.

## Conclusion

To our knowledge, this is the first report on molecular detection and characterization of DUGV in *A. variegatum and R.* (*B.*) *microplus* infesting cattle in North-Central Nigeria and provides an update on DUGV after over four decades of no information in the country. Moreover, this study indicates that DUGV-positive ticks can be found on DUGV-negative cattle suggesting that molecular studies focused on cattle only may not be enough for active surveillance. Identified genotype targets of the study differed only slightly from the first strain, which was isolated in 1964 in Nigeria.

## Acknowledgments

We are thankful to René Schöttner and Martin Abs for their excellent experimental support.

## Author Contributions

**Conceptualization:** Oluwafemi Babatunde Daodu, Albert Eisenbarth, Ansgar Schulz, Daniel Oladimeji Oluwayelu, Martin H. Groschup.

**Data curation:** Oluwafemi Babatunde Daodu, Albert Eisenbarth, Ansgar Schulz.

**Formal analysis:** Oluwafemi Babatunde Daodu, Albert Eisenbarth, Ansgar Schulz.

**Funding acquisition:** James Olukayode Olopade, Martin H. Groschup.

**Investigation:** Oluwafemi Babatunde Daodu, Albert Eisenbarth, Ansgar Schulz, Julia Hartlaub.

**Methodology:** Oluwafemi Babatunde Daodu, Albert Eisenbarth, Ansgar Schulz.

**Project administration:** James Olukayode Olopade, Martin H. Groschup.

**Resources:** Martin H. Groschup.

**Software:** Oluwafemi Babatunde Daodu, Ansgar Schulz.

**Supervision:** James Olukayode Olopade, Daniel Oladimeji Oluwayelu, Martin H. Groschup.

**Validation:** Ansgar Schulz, Martin H. Groschup.

**Visualization:** Daniel Oladimeji Oluwayelu, Martin H. Groschup.

**Writing – original draft:** Oluwafemi Babatunde Daodu, Albert Eisenbarth, Ansgar Schulz.

**Writing – review & editing:** Daniel Oladimeji Oluwayelu, Martin H. Groschup.

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
