## [Decision Letter · Decision Letter 0]

25 Aug 2021

Dear Dr. Groschup,

Thank you very much for submitting your manuscript "Molecular Detection of Dugbe Orthonairovirus in Cattle and their Infesting Ticks (Amblyomma and Rhipicephalus) in Nigeria" for consideration at PLOS Neglected Tropical Diseases. As with all papers reviewed by the journal, your manuscript was reviewed by members of the editorial board and by several independent reviewers. The reviewers appreciated the attention to an important topic. Based on the reviews, we are likely to accept this manuscript for publication, providing that you modify the manuscript according to the review recommendations. 

Sincerely,

Dennis A. Bente, D.V.M., Ph.D.

Deputy Editor

Dennis Bente

Deputy Editor

Reviewer's Responses to Questions

**Key Review Criteria Required for Acceptance?**

**Methods**

-Are the objectives of the study clearly articulated with a clear testable hypothesis stated?

-Is the study design appropriate to address the stated objectives?

-Is the population clearly described and appropriate for the hypothesis being tested?

-Is the sample size sufficient to ensure adequate power to address the hypothesis being tested?

-Were correct statistical analysis used to support conclusions?

-Are there concerns about ethical or regulatory requirements being met?

Reviewer #1: - 196bp is pretty a quite small fragment to do phylogeny. I appreciate that some Nanopore sequencing was performed, but only for one sample. If the qPCR primers amplify a very conserved region, it’s actually very hard to conclude that all your samples are closely related to the IbAr 1792 strain. And of course, no conclusion can be drawn about the diversity of the strains based on Nanopore sequencing since only one sample was used. Whether the strains are all similarly closely related to IbAr 1792 (as you said) or there is diversity in your samples, both scenarios are interesting, but needs to be better supported. Not all samples need to be sequenced but I would like to see more sequences of adequate size for reliable phylogeny, ideally from verry different samples.

- The authors mention that ticks were placed in 2ml Eppendorf tubes. Engorged Amblyomma ticks get pretty big, too big to fit in those specific tubes. Did they authors pick partially engorged Amblyomma ticks on purpose?

- Please indicate the number of the protocol approved by the Ethics committee for this work. Also clarify who performed the animal blood draws.

Reviewer #2: (No Response)

Reviewer #3: The methods were appropriate for the Dugbe virus prevalence study.

**Results**

-Does the analysis presented match the analysis plan?

-Are the results clearly and completely presented?

-Are the figures (Tables, Images) of sufficient quality for clarity?

Reviewer #1: - The authors used the term "frequency" to report the number of ticks collected. Frequency is by definition a number of occurrences PER UNIT OF TIME. It is not appropriate here.

- Please indicate the range of CT values of the samples selected for sequencing

Reviewer #2: (No Response)

Reviewer #3: The results match the presented analysis plan presented in the introduction.

**Conclusions**

-Are the conclusions supported by the data presented?

-Are the limitations of analysis clearly described?

-Do the authors discuss how these data can be helpful to advance our understanding of the topic under study?

-Is public health relevance addressed?

Reviewer #1: - It is not “extremely difficult to draw definite conclusions on vector competence based on this data”, it is impossible. Your data contributes to what is known about ticks harbouring DUGV, but there’s no way we can extrapolate on vector competence.

Reviewer #2: (No Response)

Reviewer #3: The conclusions are supported by the data presented

**Editorial and Data Presentation Modifications?**

Reviewer #1: - The running title is missing

- The author’s summary is a copy/paste from the abstract with a few scientific terms replaced. This is not what an Author’s summary is. Please look again at the guidelines.

- I would remove the estimates of ticks per cattle, simply obtained by calculating an average. When several hosts are available, ticks never infest them homogeneously. I understand that this value is sometimes helpful to compare burdens, but maybe only compare raw ratio instead of stating that “on average x ticks were found on each animal”.

Reviewer #2: (No Response)

Reviewer #3: (No Response)

**Summary and General Comments**

Reviewer #1: This manuscript by Daodu et al. provides new surveillance data on the circulation of Dugbe virus. After being first detected in Nigeria, it is very interesting to see more recent data. The manuscript is overall well written, especially the discussion section that I found of very good quality. The reason I am requesting major revision is because the authors are concluding that the strains they detected are all closely related to IbAr 1792, based on 196bp amplicons. I would like to see more sequences.

Reviewer #2: (No Response)

Reviewer #3: Overall, the study presents updated prevalence data of circulating Dugbe virus from cattle ticks. Given phylogenetic sequence analysis for the S-segment, very little variance has occurred in this virus. It would have beneficial if M-segment phylogeny work had been done to assess if the virus had undergone any reassortment events these last forty years from the previous work in the region.

PLOS authors have the option to publish the peer review history of their article (what does this mean?). If published, this will include your full peer review and any attached files.

Reviewer #1: No

Reviewer #2: No

Reviewer #3: Yes: Sergio Rodriguez

Figure Files:

Data Requirements:

Reproducibility:

References

---

## [Editor Report · Decision Letter 1]

12 Oct 2021

Dear Dr. Groschup,

We are pleased to inform you that your manuscript 'Molecular Detection of Dugbe Orthonairovirus in Cattle and their Infesting Ticks (Amblyomma and Rhipicephalus) in Nigeria' has been provisionally accepted for publication in PLOS Neglected Tropical Diseases.

Best regards,

Dennis A. Bente, D.V.M., Ph.D.

Deputy Editor

Dennis Bente

Deputy Editor

---

## [Editor Report · Acceptance letter]

25 Oct 2021

Dear Dr. Groschup,

We are delighted to inform you that your manuscript, "Molecular Detection of Dugbe Orthonairovirus in Cattle and their Infesting Ticks (Amblyomma and Rhipicephalus) in Nigeria," has been formally accepted for publication in PLOS Neglected Tropical Diseases.

Best regards,

Shaden Kamhawi

co-Editor-in-Chief

Paul Brindley

co-Editor-in-Chief
